# The Frequency of Porcine Cysticercosis and Factors Associated with *Taenia solium* Infection in the Municipality of Tuchín-Córdoba, Colombia

**DOI:** 10.3390/pathogens13040311

**Published:** 2024-04-11

**Authors:** Margarita M. Arango-Londoño, Sara López-Osorio, Fernando Rojas-Bermudéz, Jenny J. Chaparro-Gutiérrez

**Affiliations:** 1CIBAV Research Group, Veterinary Medicine School, Faculty of Agrarian Sciences, Universidad de Antioquia, UdeA, Medellín 050034, Colombia; margarita.arangol@udea.edu.co (M.M.A.-L.); sara.lopezo@udea.edu.co (S.L.-O.); 2Asociación Porkcolombia-FNP, Ceniporcino, Bogotá 111311, Colombia; frojas@porkcolombia.co

**Keywords:** backyard pigs, indigenous community, *T. solium* American/African genotype

## Abstract

Taeniasis and cysticercosis are parasitic infections that affect humans and pigs. Their global distribution constitutes a serious public health issue with significant implications for pork production. The purpose of this study was to evaluate the presence of porcine cysticercosis in backyard swine from 42 indigenous communities throughout Tuchín-Córdoba, Colombia. Between December 2020 and March 2021, free-range pigs (n = 442) were assessed using the ELISA cysticercosis Ag test; 85 pigs were examined through sublingual visual evaluation, and 4 slaughtered pig carcasses were subjected to standard operation inspection. The collected cysticercus underwent histological and PCR analysis. Furthermore, 192 surveys of knowledge, attitudes, and practices (KAP) were used to identify the factors that facilitate infection transmission. Serological investigation revealed that 9.7% (46/472) of the animals were positive for cysticerci Ag. Sublingual inspection identified cysticercus in 28.7% (25/87) of the animals, while PCR analysis indicated that cysticercus corresponded to the *T. solium* American/African genotype. The factors associated with *T. solium* infection in the pigs in the surveyed areas numbered 14. The majority are associated with factors that promote the active persistence of *Taenia solium’s* life cycle in an area, such as lack of environmental sanitation, a lack of coverage or care for drinking water and wastewater treatment services, and no solid waste disposal.

## 1. Introduction

Taeniasis/cysticercosis complex is a parasitic infection caused by parasites of the genus *Taenia* spp. and designated as a neglected disease by the WHO. This infection is extremely harmful to human and animal health, especially in tropical areas of developing African, Asian, and Latin American countries [1]. The prevalence of taeniasis/cysticercosis complex is strongly linked to poor hygienic conditions, a lack of sanitary services, poverty, eating habits, poor pig production techniques, and a lack of public health education [2]. Taeniasis is a parasitic infection caused by *T. solium* in humans. After ingesting mature cysticercus (larvae stage) found in muscle and various tissues of infected pigs, humans become infected. Tapeworms stick to the intestinal mucosa, mature, develop proglottids, become gravid, and release eggs in the feces [3]. On the other hand, both humans and pigs can contract cysticercosis [4], which is caused by the ingestion of *T. solium* eggs found in food or water contaminated with human feces through the fecal–oral route. Eggs hatch and, after passing through the stomach, release an embryo that travels through the bloodstream and infects various parts of the body, primarily the muscles and nervous system [5].

The WHO and WHOA have designated the disease as potentially eradicable based on its prevalence and importance to public health and have set goals for 2030 that are linked to a roadmap focused on reducing risk factors linked to the presence of the disease, allowing prevention, elimination, and eradication [1,6,7]. Colombia is one of the countries where the disease is considered to be endemic, but there are few studies on the state of the infection in the country [8,9], where a seroprevalence of around 8.5% was recorded in the general human population in 2010, with a more pronounced distribution in the north and south of the country associated with socioeconomic factors and sanitary and cultural conditions [10]. As a result, studies that allow an updated approximation of the epidemiological condition of the infection in the national territory are required, allowing the design of comprehensive strategies for the control, elimination, and eradication of the taeniasis/cysticercosis complex.

In Colombia, there were 5,950,113 pigs counted nationally in 2021, of which 1,374,296 were found to be produced in backyards [11]. It is important to note that backyard production continues in remote areas where there is still a general lack of pig slaughterhouse facilities and pork inspection, and control is poor or non-existent. As a result, it is critical to conduct epidemiological studies to identify areas where the *Taenia solium* life cycle is present and active, which generally coincide with areas with poor environmental and social conditions [12,13].

To achieve an initial approximation, we proposed to investigate the frequency of porcine cysticercosis and the molecular identification of *Taenia* spp. along with factors associated with its presentation in the indigenous communities from the municipality of Tuchín-Córdoba. As a contribution of baseline information for the design of strategies for a control, elimination, and eradication program of the taeniasis/cysticercosis complex in Colombia, possible work roadmaps were defined in the short–medium and long term following the recommendations of the WHO and as part of the National Technical board for the reduction in the taeniasis/cysticercosis complex of the Ministry of Health and Social Protection.

## 2. Materials and Methods

### 2.1. Study Area

The study area was selected due to an observed free-range pig husbandry and anecdotal information on the presence of cysticercosis by veterinary technicians that worked in the area. In addition, no previous study on porcine cysticercosis has been conducted in the zone. Tuchín is a municipality in the Colombian state of Córdoba, located at 9°11′09″ N 75°33′19″O/9.1858333333333, −75.555277777778, with an area of 128 km^2^. It has an elevation of 106 m above sea level, an average temperature of 28 °C, and a population of 40,033 inhabitants, of which 39,101 are indigenous descendants of the Zenú ethnic group, distributed in 65 communities [14]. Agriculture and handicrafts are among the main economic activities of the region, with this population dedicated to the fabric of the iconic Colombian hat *vueltiao*. Livestock, fishing, and pig and poultry farming are also important for subsistence.

### 2.2. Sample Size

In the years 2020–2021, the study was conducted in 42 indigenous villages in Tuchín-Córdoba selected by convenience considering the location and acceptance to participate in this study. The communities of the municipality’s reservation served as the research unit, and the sample size was calculated using the municipality’s Beneficiary Selection System for Social Programs (SISBEN). The selection of the sample was convenient, and only 192 households agreed to participate in the study. According to the ICA (Colombian Agricultural Institute) [15] porcine census, Tuchín had 11,322 pigs in 2019. The sample size for the detection of antigens of cysticercosis was 472 pigs. The sample size calculation was made for finite populations with a confidence of 95%, error of 3%, and a known prevalence of 13.33% [16] using the formula n = Z^2^PQ/L^2^ [17].

### 2.3. Sublingual Inspection

Eighty-seven pigs were sublingually inspected. These pigs were selected according to the consent of the owner to make the inspection. We were not allowed to perform the lingual inspection on all the sampled pigs. Each pig was briefly immobilized with a noose, then a wooden stick was used to open and hold the animal’s mouth open, and then the tongue was carefully moved and visually examined and palpated, particularly the ventral aspect and near the base of the tongue. Cysticercosis was diagnosed in all pigs with one or more cysts on the tongue or tongue base [18].

### 2.4. ELISA

Blood samples were taken from 472 pigs raised on backyard farms at various phases of fattening. Approximately 5 mL of blood was taken from each pig. Breeding males and females from 4 months onwards and pigs that were difficult to manage or aggressive were excluded. We used 21-gauge needles and BD Vacutainer^®^ (Auckland, New Zealand) serum tubes with separating gel to bleed the pigs. The extracted samples were centrifuged for five minutes at 1200× *g* to obtain the serum. The sera were kept at −20 °C in Eppendorf tubes for further analysis.

The titers of *T. solium* cysticerci antigens in the pig sera were determined using the commercial kit for cysticercosis (Ag-ELISA apDia^®^ Ref. 650510 Turnhout, Belgium), according to the manufacturer’s instructions. Duplicates of controls and samples were processed following the manufacturer’s protocol. The absorbance values/optic densities (ODs) were determined at 450 nm within 15 min of stopping the reaction with 0.5 M H_2_SO_4_. An ELISA plate washer (Thermo Scientific well wash -N10800-05 19111745) was used, as well as an EpochTM Microplate Spectrophotometer (Biotek Instruments, Inc., Winooski, VT 05404-0998, USA). The Gen5 program was used to process the data.

### 2.5. Postmortem Inspection

Due to the lack of slaughterhouses in the area, the animals were slaughtered in houses set aside for this purpose within the indigenous community. This was carried out using the electrocution method. Four pig carcasses were purchased from communities where pigs with positive sublingual inspection were observed and then examined in accordance with the OMSA Terrestrial Manual [19] and Chembensofu [20]. The number of carcasses was determined by convenience due to the lack of resources. To ensure traceability, the sections of the carcass were labeled with a specific code immediately after slaughter. The carcass was then separated into two halves, starting with the caudal skull, which was divided into equal halves, and the head was removed by cutting through the neck vertebrae. The organs were then separated from the carcass, and inspection of cysts in the abdominal cavity was carried out. After that, half of the carcass was deboned, removing and retaining as much muscle tissue as possible while keeping the masseter muscles and muscles of the base of the tongue marked and separated from other skeletal muscles, and all connective and fatty tissue was removed by looking for cysts using thin sections (5 mm) of the muscles and organs in a systematic manner, paying attention to between the layers of muscle, fat, and skin.

In addition, the existence of *Taenia* spp. cysticerci was determined by visual detection and palpation of cysts in the heart, tongue, masseter muscles, esophagus, diaphragm, eyes, brain, and liver. The total number of cysts (sum of cysts from each estimated total muscle group plus counts of cysts in the masticatory muscles and other organs) was calculated, and the cysticerci stage was classified as viable, degenerate, or calcified according to Boa [21]. Furthermore, if cysts were discovered, they were kept in Eppendorf tubes or Falcon tubes with 70% ethanol for multiplex PCR analysis, and some samples were fixed in 10% buffered formaldehyde for histological study.

### 2.6. Histopathological Analysis

The cysticerci were fixed in 10% buffered formaldehyde and kept at room temperature before being processed in the University of Antioquia’s Animal Pathology Laboratory, where they were embedded in paraffin and cut into 3 µm tissue sections to be processed with routine Hematoxylin–Eosin stain to perform a descriptive histopathological analysis of the cyst. The CellSens Standard software Version 4.2 (Olympus Corporation, Center Valley, PA, USA) was used to take multiple photomicrographs of the cysticerci using an. Olympus BX53 light microscope with an Olympus DP74 digital camera (Olympus Corporation, Center Valley, PA, USA).

### 2.7. DNA Extraction and PCR

At least two cysticerci obtained previously from each carcass were macerated individually, and the parasite’s DNA was extracted using the DNeasy Blood and Tissue kit (Qiagen^®^, Cat. No. 69504, Hilden, Germany), according to the manufacturer’s instructions. To differentiate the species and genotypes of *Taenia* spp., a multiplex PCR was used [22], based on the nucleotide sequences of *Cox1* [Table 1]. A volume of 25 µL of reaction mixture was used for amplification with the Hot Start Taq Master Mix kit (Qiagen^®^, Cat. No.203446). Electrophoresis was performed on the PCR products in 1% agarose gel for 55 min at 90 V. Positive controls were donated by the Institute of Parasitology Justus Liebig University, Giessen. Sequence of specific primers and the conditions used in the multiplex PCR are described in Table 1.

### 2.8. Household Questionnaire

A structured questionnaire with two sections was administered through personal interviews with a member of the selected household who was familiar with the day-to-day raising of pigs and the relevant knowledge, attitudes, and practices (KAP): (1) general information about household characteristics, such as age, gender, ethnicity, education level, household size and toilet type used, source of water, hygiene practices, and respondents’ knowledge and perceptions about the taeniasis/cysticercosis complex, and (2) information about pig management, such as herd size, pig raising system, animal health practices, pig origin, pig sale and slaughter, and proportion of own pigs consumed. A total of 80 variables were evaluated in the questionnaire. A representative of the 192 households included answered the questionnaire, although the questions concerning pigs were only answered by 179 people, since 13 of the 192 households had no pigs at the time of the study. 

### 2.9. Statistical Analysis

Descriptive statistics, mean, standard deviation, and the relative frequency of positive cases by community, number, and cysticercosis stage were utilized. The 95% confidence interval for frequency was calculated using binomial calculation. The odds ratio (OR) of the variables was assessed using multiple correspondence analysis to determine factors associated with cysticercosis. All survey data were entered into sheets on Microsoft^®^ Excel^®^ 2019 MSO (16.0.10386.20017) 64-bit calculation tool for tabulation and processed in R Studio [23] v. 4.4.1.2. The library also made use of R Studio’s FactomineR to perform multiple correspondence analysis.

## 3. Results

### 3.1. General Description of Pig Backyards

A total of 472 pigs from 179 families were examined. Most of the pigs sampled were a mix of Colombian creole pig breeds (black pigs and casco de mula); they were predominantly females (59.1% (279/472)); 65.8% were over 12 months of age (311/472); and 98.8% (177/179) of the pigs were raised in a free-range environment. About 51.3% (92/179) of the pigs were fed with cereals, cassava, yams, and harvest byproducts, while 48.6% (87/179) were fed with kitchen scraps. The source of water for the pigs was lakes, streams, or rainwater. None of the owners reported receiving veterinary care while raising the pigs. Only a small percentage of owners reported deworming their pigs (9.5% (17/179)). All families raised pigs for both sale and consumption; 21.7% (39/179) of owners slaughtered and sold pigs at home; and 90.5% (162/179) had their pigs slaughtered without inspection.

### 3.2. General Description of Population

Elementary school was the level of education for 60.9% (117/192), 22.9% (44/192) were illiterate, 14.6% (28/192) had a high school education, and 1.6% (3/192) had a technical school education. In total, 53.6% (103/192) of the families lacked sanitary facilities, 32.2% (62/192) used a peasant sanitary bowl, and 14.1% (27/192) had pit latrines; 88.8% (159/179) of owners said that the animals had access to human excrement. Only 32.2% (62/192) reported having access to a septic tank; however, 93.2% (179/192) said that their family members frequently engaged in outdoor defecation. For human consumption, every home used rainfall, water from lakes, and water from streams. *Taenia solium* was unknown to 52.1% (100/192), and 54.1% (104/192) did not know how it is transmitted. Hygiene was poor; only 31.8% (61/192) washed their hands regularly after toilet use. A total of 20.8% (40/192) had eaten pork with visible cysticercus, 30.7% (59/192) had given pork with visible cysticercus to other members of the community, 5.7% (11/192) had sold a pig with visible cysticercus, and 4.7% (9/192) had a history of epilepsy in the household.

### 3.3. Frequency of Porcine Cysticercosis

The frequency of porcine cysticercosis in this study was 9.7% (46/472), determined by Ag-ELISA in 472 serum samples. Cysticerci were found sublingual in 28.7% of the examined pigs (25/87) (Table 2). In addition, all four pigs examined by postmortem inspection tested positive for viable cysticerci with high infection levels of >100 cysticerci, determined according to Phiri [24]. A total of 304 cysticerci were found in common predilection sites such as the masseter, fore and hind limb muscles, and the psoas. Of these, 57.2% (174/304) were viable cysticerci, 32.8% (100/304) were degenerated, and 9.8% (30/304) were calcified. Microscopic examination revealed the cross section of a larva with invaginated scolex, suckers, and hooklets. Amplification of the *Cox1* gene (720 bp) corresponding to the *T. solium* American/African genotype was found in the cysticerci of the four animals examined postmortem by conventional PCR. Amplification of *T. asiatica* (269pb) or *T. solium* genotype Asiatica (984pb) by PCR multiplex was not observed.

### 3.4. Risk Factors for Porcine T. solium Cysticercosis

In the evaluation of the factors associated with a diagnosis of cysticercosis, 33 significant variables were obtained with an OR greater than 1 (*p* < 0.05). To see the probability distribution and contrast the hypotheses, a Chi square test of all variables against positive diagnosis in the ELISA test for cysticercosis found 14 significant variables [Table 3].

## 4. Discussion

Although cysticercosis is considered endemic in Colombia [25], there are a lack of up-to-date data on its prevalence, which is required for the design and implementation of a program for the control, elimination, and eradication of the taeniasis/cysticercosis complex in accordance with the WHO roadmap [26,27,28]. In the current study, we aimed to assess the frequency of cysticercosis and factors related to pigs in the municipality of Tuchín in the department of Córdoba. We discovered a frequency of porcine cysticercosis of 9.7% (46/472), determined by Ag-ELISA in serum. Circulating antigens can be detected between 2 and 6 weeks post-infection and can be detected for up to 6 months, even in pigs with a low-rate infection (one cyst) [29,30]. Cysticercosis Ag-ELISA only determines the presence of viable cysticerci [31,32], unlike antibody detection, and the antigen levels are associated with parasite burden [33,34]. The low sensitivity of the Ag-ELISA test in detecting low-burden cysticercosis infection has been described before [20,35]. These studies reveal unsatisfactory sensitivity in pigs with a mild cyst load, which restricts its application as a general screening tool for pig diagnosis or control. If we compare the Ag frequency (9.7%, 46/472) with the sublingual inspection (28.73%, 25/87), there is a great difference, which could be due to various reasons. The sublingual inspection is a diagnostic technique with a low sensitivity of 16–70% that varies depending on the severity of the illness and a high specificity close to 100 percent [29,31,32]. Although it is a cheap strategy, it is challenging to use on backyard pigs who are not used to human contact and handling. That is why, for the sublingual examination, only 85 of the 472 pigs who were serologically sampled were allowed to be tested; as a result, although the veterinarian tried to conduct the sublingual examination on all the pigs, only the reduced number were successfully evaluated in a relatively short period of time. In the case of the other animals, the owner asked for immediate release after the second or third attempt. Furthermore, 48.0% (12/25) of the animals with a positive sublingual cysticercosis inspection had a positive Ag-ELISA test. An explanation could be related to the level of infection, the immunological response of the pigs after infection, and the number of viable cysts present in the animals. The examination of the tongue of the animal by visual inspection and palpation detects only the most heavily infected pigs [18]; and following experimental *T. solium* infection in pigs, it was found that the number of viable cysts recovered at necropsy can help to explain variation in antigen concentration in serum, and that the decrease in cyst numbers in older animals may be due to their more potent innate and acquired immune responses [30,35,36].

This frequency is comparable to that found in neighboring municipalities of the region, Moñitos and Los Córdobas, where postmortem inspection revealed a prevalence of 13.33% [13], ranging from 0.25% in the central region or middle Sinu to 22.2% in municipalities of the coastal zone and Savannah region, specifically Moñitos, Ciénaga de Oro, Chinu, and Sahagun. The serofrequency found was also similar to that reported in other countries such as Brazil with a seroprevalence of 5.3% of porcine cysticercosis in rural communities in eastern Minas Gerais [37], 9.01% in the Sierra of Northern Ecuador [38], and 10.48% in Myanmar in the Nay Pyi Taw area [39].

Given that full carcass dissection is the gold-standard diagnosis method, the fact that more than half of the cysts identified (57.4%) were viable is cause for concern. Cysticerci were recovered primarily from psoas and the muscles of the forelimbs and hindlimbs. Other studies reported similar findings [21,24,40,41]. The high number of viable cysticerci in the back and hind legs was also recently reported in [40,41]; and it is a concern that is not always part of the standard meat inspection protocol. Although the masseters, tongue, and heart are tissues and organs that are recommended for routine meat inspection for cysticercosis, they showed a low or absent cyst count in this study. Even though the four carcasses examined had a significant number of cysticerci, only one of them had cysticerci discovered in the tongue, and two showed cysticerci in the masseters and heart after making small cuts. These findings support those reported by Boa [21] and da Silva [42], who discovered that only 10.6% and 18.06% of metacestodes, respectively, were present in the muscles or organs used for identifying cysticercosis. Previously, Lightowlers [43] reported that the inclusion of the muscles from the right or left forelegs together with the heart, tongue, and masticatory muscles raised the diagnostic sensitivity to 83–88% in a partial carcass dissection. Moreover, it is important to consider the difficulty for this study to obtain animals for full carcass dissection, since the owners only slaughtered animals for festivities, the cost of purchasing animals is high, and some owners prefer to sell them to other members of the community or, in some cases, to consume themselves. Other authors have reported additional limitations of this technique such as the time required for slicing muscle tissue to identify and count cysts, and the need for trained personnel to perform this procedure [43].

There are no prior publications that identify *Taenia* spp. infecting pigs from Colombia, and the Ag-ELISA does not distinguish this aspect because the assay shows cross-reactions with other *Taenia* species [29]. Molecular techniques for the identification of *T. asiatica* and Asian versus American/African genotypes of *T. solium* were crucial in this study. By using a multiplex PCR based on primers to amplify *Cox1*’s gene nucleotide sequences, the American/African genotype of *T. solium* was discovered in all cysticerci examined. These results are in accordance with the findings of [44,45,46] in surveys of porcine cysticercosis.

The results of the serology and sublingual inspection, along with the examination of the four postmortem carcasses with high infection levels and the PCR confirmation of the species, indicate that the pigs in Tuchín municipality were exposed to *T. solium* eggs. Most households (93.2% (179/192)) indicated that their family members frequently engaged in outdoor defecation, and while some families used pit latrines or peasant sanitary bowls, just 32.2% (62/192) of interviewees said they had access to a septic tank, and 88.8% (159/179) of pig owners said that the animals had access to human excrement. The present study reports the following associated factors to find cysticerci-seropositive pigs: human outdoor defecation (OR = 1.82; 95% CI = 0.97–3.63), pigs raised in a free-range environment (OR = 2.31; 95% CI = 1.12–4.75), pigs have access to consume human feces (OR = 3.36; 95% CI = 1.38–8.19), no use of antiparasitic treatment for pigs (OR = 10.73; 95% CI = 3.25–35.33), illiterate owners (OR = 15.00; 95% CI = 5.24–42.91), and the animals that die are destined for consumption by other animals (OR = 5.81; 95% CI = 2.23–15.12). Sarti [46] in Mexico claims that the expansion of *T. solium* in rural areas is aided by a lack of understanding of the parasite life cycle and socioeconomic factors, like sanitation, pig husbandry, and pig contact with human feces, affecting transmission. The results of this study are also in line with earlier reports from Diaz [47] in Peru, Shey-Njila [48] in Cameroon, Ngowi [49] in Tanzania, Krecek [50] in South Africa, Eshitera [51] in Kenia, and Acevedo-Nieto [37] in Brazil, who found a higher seroprevalence of porcine cysticercosis in households without latrines and in households with pigs that were not permanently confined and could gain access to human feces.

Other risk factors of porcine cysticercosis found in this study are slaughtering pigs at home (OR = 4.17; 95% CI = 1.80–9.65), selling the pigs at home (OR = 3.11; 95% CI = 1.48–6.52), a lack of knowledge of the transmission of porcine cysticercosis (OR = 5.46; 95% CI = 1.79–16.62), the consumption of pork with cysticerci (OR = 2.69; 95% CI = 1.20–6.04), the sale of pigs with cysticerci (OR = 3.23; 95% CI = 1.37–7.61), and not knowing the consequences of eating pork with cysticerci (OR = 3.63; 95% CI = 1.84–7.17). These results are coincident with the previous reports of Boa [52] in Tanzania and Sikasunge [53] in Zambia, who noted that home pig slaughter and the absence of pork inspection were risk factors for human taeniasis.

Additionally, findings that are similar to those of our study have been reported in other African communities where people have eaten and sold pork that was contaminated with cysticerci, as well as in situations where people have sold cysticerci-infected pigs to other community members rather than eating them [53].

In our study, unsanitary practices such as not washing hands before and after handling food (OR = 3.40; 95% CI = 1.76–6.59) and not washing hands after defecation (OR = 5.86; 95% CI = 2.98–11.51) were identified as risk factors for porcine cysticercosis. Furthermore, we found that the community did not have access to potable water, so they drank unsafe water. These variables have previously been identified as risk factors for human *T. solium* infection in Burkina Faso [54]. Although we did not evaluate the human infection, the presence of porcine cysticercosis and all the characteristics described for this municipality indicate that the taeniasis/cysticercosis complex is present and active. Furthermore, a study should be conducted to determine the risk factors associated with prevalence among the various communities, as a difference in the proportion of the 13 positive communities of the 42 evaluated, because membership to specific communities was identified in this study as a risk factor for swine cysticercosis.

Because official inspection reports in slaughterhouses registered to the national authority with technologically advanced porcine production systems show that porcine cysticercosis is not present (INVIMA), studies in different regions of Colombia are still needed to approximate the epidemiological state of the disease in the country’s pig population.

This study provides updated information to begin the design of strategies for a program to control, eliminate, and eradicate the taeniasis/cysticercosis complex in this region of Colombia.

## 5. Summary

Porcine cysticercosis is a serious public health and economic concern caused by the zoonotic tapeworm *T. solium*. A complex network of biological and social factors maintains its endemic status and limits success in the disease`s control. Free-range pigs are obligate intermediate hosts, making them prime targets for local control and pilot studies. In Colombia, the disease is considered endemic, but there are no current data on the infection’s state in the country, where a seroprevalence of around 8.5% was recorded in the general human population, with a more pronounced distribution in the north and south of the country associated with socioeconomic factors and sanitary and cultural conditions. We studied the frequency of porcine cysticercosis, the molecular identification of *Taenia solium*, and the factors associated with its presence in indigenous communities in the municipality of Tuchín-Córdoba. This municipality possesses all the characteristics that make it vulnerable to the presence of the taeniasis/cysticercosis complex, including inadequate basic environmental sanitation, a lack of coverage or care for drinking water and wastewater treatment services, and no solid waste disposal. This study provides updated information to design strategies for a control program to eliminate and eradicate the taeniasis/cysticercosis complex in this region of Colombia, and serves as a pilot study for the activation of interventions by Colombia’s National–Intersectoral Board for the Elimination of the taeniasis/cysticercosis complex, led by the Ministry of Health and Social Protection.

## Figures and Tables

**Table 1 pathogens-13-00311-t001:** Primers and PCR conditions for *Taenia* spp. PCR.

Name	Sequence	Size (bp)	PCR Conditions
*T. asiatica*	5′-ACGGTTGGATTAGATGTTAAGACTA-3	269	35 cycles: 94 °C 30 s. 60 °C 30 s. 72 °C 90 s.
*T. solium* American/African genotype	5′-GGTAGATTTTTTAATGTTTTCTTTA-3′,	720	
*T. solium* Asian genotype	5′-TTGTTATAAATTTTTGATTACTAAC-3	984	
Common reverse primer	Rev, 5′-GACATAACATAATGAAAATG-3		

**Table 2 pathogens-13-00311-t002:** Frequency of porcine cysticercosis by sublingual and Ag-ELISA test.

Community Number	Sublingual Inspection				Ag-ELISA			
	Pigs (n)	Positive Pigs	Frequency (%)	95% CI	Pigs (n)	Positive Pigs	Frequency (%)	95% CI
1	9	2	22.2	(2.8–60)	10	1	10	(0.2–44.5)
3	6	1	16.6	(0.4–64.12)	26	2	7.7	(0.9–25.1)
4	5	2	40	(5.3–85.3)	6	1	16.7	(0.4–64.1)
7	3	1	33.3	(0.8–90.6)	16	1	6.2	(0.2–30.2)
8	7	2	28.6	(3.7–7.9)	34	3	8.8	(1.8–23.7)
9	9	3	33.3	(4.5–70.1)	38	6	15.8	(6.0–31.2)
15	8	2	25	(3.2–65.1)	18	3	16.7	(3.6–41.4)
16	11	5	45.5	(16.7–76.6)	24	13	54.2	(32.8–74.4)
18	6	1	16.7	(0.4–64.1)	10	2	20	(2.5–55.7)
19	8	1	12.5	(0.3–52.6)	20	4	20	(5.7–43.7)
23	7	3	42.9	(9.9–81.6)	18	7	38.9	(17.3–64.2)
27	3	1	33.3	(0.8–90.6)	8	2	25	(3.2–65.1)
29	5	1	20	(0.5–71.6)	18	1	5.6	(0.1–27.3)
Negative ^1^	0	0	0		226	0	0	
Total	87	25	28.7	(19.4–39.4)	472	46	9.7	(7.2–12.8)

^1^ Negative community numbers (2-5-6-10-11-12-13-14-17-20-21-22-24-25-26-28-30-31-32-33-34-35-36-37-38-39-40-41-42). The frequency was calculated per community due to the difference in the sampled pigs in each community.

**Table 3 pathogens-13-00311-t003:** Variables associated with positive results in the ELISA test for cysticercosis.

Factor	Level	*n*	Positive Case	Negative Case	Odd Ratio	*p* Value
Pigs raised in a free-range environment	Yes	259	33	226	2.315 (1.128–4.754)	0.01
	No	213	13	200		
Consumption of pork with visible cysticerci	Yes	51	11	40	2.696 (1.202–6.045)	0.01
	No	421	35	386		
Commercialized the pigs at home	Yes	242	31	211	3.113 (1.485–6.526)	0.00
	No	230	15	215		
Sale of pigs with cysticercosis	Yes	39	8	31	3.234 (1.373–7.616)	0.00
	No	433	38	395		
Pigs had access to consume human feces	Yes	305	40	265	3.367 (1.383–8.194)	0.00
	No	167	6	161		
No washing of hands before and after handling food	Yes	118	21	97	3.408 (1.763–6.590)	0.00
	No	354	25	329		
Did not know the consequences of eating pork with cysticerci	Yes	172	29	143	3.638 (1.844–7.178)	0.00
	No	300	17	283		
Lack of knowledge of transmission of *Taenia solium*	Yes	339	42	297	3.832 (1.336–10.986)	0.00
	No	133	4	129		
Slaughtered pigs at home	Yes	262	34	228	4.179 (1.809–9.653)	0.00
	No	210	12	198		
	No	451	34	417		
Lack of knowledge of transmission of porcine cysticercosis	Yes	16	5	11	5.468 (1.798–16.623)	0.00
	No	456	41	415		
	No	466	38	428		
The pigs that die are destined for consumption by other animals	Yes	271	41	230	5.814 (2.235–15.123)	0.00
	No	201	5	196		
No habit of washing hands after defecation	Yes	112	24	88	5.864 (2.987–11.511)	0.00
	No	360	22	338		
Use of antiparasitic treatments in pigs	Yes	204	9	195	10.732 (3.259–35.334)	0.00
	No	268	37	231		
Illiterate	Yes	198	41	157	15.000 (5.243–42.916)	0.00
	No	274	5	269		

## Data Availability

The raw/processed data required to reproduce the above findings cannot be shared at this time due to legal/ ethical reasons.

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
