# Peer review of "The Frequency of Porcine Cysticercosis and Factors Associated with Taenia solium Infection in the Municipality of Tuchín-Córdoba, Colombia"

_pathogens, 2024, doi:10.3390/pathogens13040311_

Round 1

Reviewer 1 Report

Comments and Suggestions for Authors

This is a descriptive study on the prevalence of cysticercosis in free range pigs, carried out in rural indigenous communities, in a poor and isolated region in the Province of Cordoba in Colombia. Infection of the pigs was detected through immunological testing (ELISA) and sublingual inspection; all cysts’ isolates were African/American genotype, as determined by PCR. The work was well designed, methodologies used were adequate, the manuscript is reasonably well written and requires minor English language correction. The study also includes household surveys on the population to identify the social determinants promoting transmission of the parasite.

However, this report contains no novel/original information on the parasite or the disease, except that this region in Colombia had not been studied. Even the risk factors promoting transmission of taeniasis cysticercosis such as lack of veterinary inspection, deficient hygiene, lack of education, etc., are well known through many similar field studies. Identification of adult tapeworm carriers in the indigenous population would provide more interest to the study. In summary, this is a useful piece of information for local authorities in charge of the surveillance and prevention of taeniasis/cysticercosis in Colombia but are of reduced interest for the expert reader.

Before considering publication of this manuscript I recommend a major revision, including description of primers in Table 1 as part of the text; figures 1, 2 and 3 should be deleted or included as supplementary information; figure 4 should also go supplementary information.

Comments on the Quality of English Language

Manuscript is reasonably well written and requires minor English language correction.

Author Response

Please find attached the letter

Reviewer 2 Report

Comments and Suggestions for Authors

The present paper was designed to investigate the frequency of porcine cysticercosis in backyards swine from 42 indigenous communities and factors associated with Taenia solium infection in the municipality of Tuchín-Córdoba, Colombia. To this end, the animals were subjected to a serological test (442 samples) by ELISA Cysticercosis Ag test, while the search for sublingual cysticercus was carried out on 85 pigs. In addition, cysticercus collected from four slaughtered pig carcasses were subjected to routine histopathology and PCR analysis and sequencing for the molecular characterization of T. solium genotype. Socio-demographic, behavioral and knowledge data collection, including common risk factors associated with human and porcine cysticercosis, including knowledge about taeniasis and human cysticercosis, knowledge of porcine cysticercosis, practices about pigs breeding and pork consumption, environmental conditions, hygiene practices, and housing quality were obtained using an individual questionnaire. In addition to the description of reported swine and human cysticercosis, the authors could report about teniasis in Colombia. The manuscript is scientifically sound with a significant contribution given the scarcity of published material. As below reported, the methodology requires further detailed information to allow its reproducibility. In addition, the authors did not clearly emphasize the limitations of their study.

Specific comments:

1. Page 2- Introduction section

Lines 62, 152,153 - why the molecular identification of Taenia at species level?  Was there any chance of pigs being colonized by genotypes from different continents? 

2. Line 89 - What does ICA stand for?

3. Page 3.

Lines 95-100 - Text would be better placed at beginning of 2.4 (ELISA)

Lines 150-158 - How sequencing and phylogenetic analysis was performed?

4. Line 118 - In what way were the pigs sacrificed?

Page 7- Results section

5. Lines 213-215 - Figure 2 does not describe these morphological findings.

6. Line 219 - Amplification of T. asiatica (269pb) or T. solium genotype Asiatica (984pb) by PCR multiplex was not observed (data not shown).

Isn't that an expected result? Pigs were brought into Latin American countries by Spanish and Portuguese colonists.

The sequences used in the analysis were retrieved from the GenBank database? Which access numbers?

7. Line 225 - Reverse the presentation order: B before C. In figure B, the cysticercus should be marked with an arrow for easier identification.

8. Lines 227-230 - Figure 2- Authors might mention what they are trying to show with figure A. Other formation: (H and I) detailed cyst in muscular tissue. (J) isolated muscular cyst. (K and L) Cyst after removal from tissue.

9. Lines 232-235 - Figure 3. Figure 3. The indication of the figure is repeated.

Histopathological detail of the cyst.

The images shown (A and B) do not match the figure title. I suggest: Cysticercus gross appearance (A and B) fixed with formaldehyde.

(C and D) cyst - microscopic examination revealed the cross section of a larva with evaginated scolex, suckers (black arrow) and hooklets (blue arrow).

The latter information is not available.

10. Line 236 – Assosiated - Typing error

11. Reverse the presentation order: Table 3 before Figure 4

12. Page 9 – Line 241 Table title - ….. for ELISA for porcine cysticercosis

13. Page 11- table 3 - What does * stand for?  

Page 12 – Discussion section

14. Lines 264-266.  Although it is a cheap strategy, it is challenging to use on backyard pigs who are not used to human contact. That’s why, for sublingual examination, only 85 of the 472 pigs who were serologically sampled were allowed to be tested. 

It seems that the inspection or cysticercus on the tongue was a request from the owners of the animals, as highlighted in the line 102.

15. Page 13-

Line 314 - The word “specie” doesn’t exist in biology.

16. Lines 363-365 - This proposal was also recently nominated. See: Acta Tropica 242 (2023) 106907.

Author Response

Dear Reviewer

Please find attached the letter

Thanks

Reviewer 3 Report

Comments and Suggestions for Authors

This manuscript provides an interesting contribution to “frequency” of Cysticercosis in backyards swine in Colombia using ELISA and PCR. Risk factors for the diseases have also been considered.

I believe that the manuscript can be enriched if the following suggestions are considered:

Major

L239-240, because all variables were tested vs ELISA test and not by sublingual inspection.

Table 3

The human outdoor defecation variable does not show a significant association as the lower value of the CI is less than 1.

L1, L252, the concept of frequency used in the document is wrong, you determined the prevalence of the disease.

L407, The reference list is more than 5 years old, it needs to be updated.

Minor

L21-24. This sentence needs to be restructured as it is long and uses a lot of connectors.

L33, This word (poor) is not appropriate for this sentence

L114, replace H2SO4 by H2SO4

L195, replace 1.6 percent by 1.6%

L183, delete “unknown breed”, as it further describes

L197, remove the full stop after latrines. Idem L215, calcified

L219, L256, insert spacing between words

L227, replace sacrifice by slaughter

Table 3

There are only 32 variables in the table.

Use full stop, not a comma, for numerical values

References

L249, eliminating and eradication, do they mean the same?

L294, This word “made” is not appropriate for this sentence

Comments on the Quality of English Language

Minor editing of English language required

Author Response

(The authors gave the same response as above.)

Reviewer 4 Report

Comments and Suggestions for Authors

This manuscript by Arango-Londono MM and colleagues describes prevalence and risk factors of Teania solium infections in pigs in 42 indigenous communities in Tuchin-Cordoba, Colombia.  The authors tested 442 pigs by cysticercus Ag, 85 pigs by sublingual visual evaluation and four pigs by necropsy followed by standard operation inspection.  The prevalence of T. solium infection by the first two methods were 9.7% (46/472) and 28.7% (25/87), respectively.  They had found many risk factors ranging from general sanitation, pig farming to education and knowledge.  It is an interesting discovery that will have practical applications on control and management and eventual eradication of this parasitic infection in pigs and humans in Colombia.

The main concern is the number of questionnaire.  In total, 179 families were included in the study, and ONE member of each household was interviewed according to M&M.  Therefore, the number should be 179.  Nevertheless, the number used in the section 3.2 between lines 193 and 206 is 192 except one in line 197, which is 179.  Please explain in the text that total number should be 192 instead of 179, so readers will have better understanding.  In line 316, 192 is used as household instead of 179.  Which one is correct?

Figure 4: there are two bands in the lane labelled C3-2, the top band is the right size, the bottom one is approximately 500bp.  Please discuss what it might be in discussion.

Table 3: please use a period “.” rather than a comma “,” to show decimals.  Please also keep only two decimals in the column labelled “Odds ratio” to be consistent with the mani text.  What do you mean by “Unlettered”, illiterate?

L267-8: “Furthermore, only 14.1% (12/85) of the animals with a positive sublingual cysticercosis inspection had a positive Ag-ELISA test”.  This statement is incorrect.  Eighty-even pigs were visually examined for cysticercus sublingually, 25 were found positive.  The rate should be 48.0% if 12 were found cysticercus Ag positive (12/25) unless I misunderstood what you meant here.

M&M section: use centrifuge force (×g) rather than rpm since rpm is meaningless without the diameter of the rotor used. Also please add information for each supplier such as CellSens, Olympus microscope and DP74 camera

Subtitle for the section 3.4.- considering change it to “Risk factor…”.

Minor points:

L18: “9.7%” rather than 10% (46/472).  Do not round up for consistence.

L19: Cysticercoid is different from cysticercus.  Make sure what you mean cysticercus here.

L28: genus, not genera.

L76: Use same format for geographical locations; “128 km2

L81: Livestock since the word start a sentence.

L85: What is “ubication”?

L90: “detection of antibodies” should be “detection of antigens”

L114: “H2SO4
L153: “a multiplex PCR was used”

Table 1: Seconds should be shortened as “s”, not seq.

Fig 1 legend: LL225: reverse the order of “B” and “C”.
Fig 2: Add letter A to the first panel; L230: (K) should be (L).

Fig 3: clearly mark the objects with descriptive arrows, which are currently missing.

L286: “psoas”

L293: “heart”

L308: “Identification”

Comments on the Quality of English Language

Many corrections are suggested. 

Author Response

(The authors gave the same response as above.)

Round 2

Reviewer 1 Report

Comments and Suggestions for Authors

The authors have followed most of my previous recommendatiosn. 

In my opinion the manuscript can be accepted in its present form.

Author Response

We appreciate all the reviewer's suggestions and edits, which have helped the manuscript become better.

Jenny J. Chaparro-Gutierrez

Reviewer 3 Report

Comments and Suggestions for Authors

The authors have responded to each of the reviewers´ comments and suggestions.

Author Response

Thank you very much for your insightful comments on how to improve the manuscript's quality.

Sincerely,

Jenny J. Chaparro-Gutierrez

Reviewer 4 Report

Comments and Suggestions for Authors

The authors have not addressed any main concerns of this review as showed below (copied and pasted from the comments on V1):

The main concern is the number of questionnaire.  In total, 179 families were included in the study, and ONE member of each household was interviewed according to M&M.  Therefore, the number should be 179.  Nevertheless, the number used in the section 3.2 between lines 193 and 206 is 192 except one in line 197, which is 179.  Please explain in the text that total number should be 192 instead of 179, so readers will have better understanding.  In line 316, 192 is used as household instead of 179.  Which one is correct?

Figure 4: there are two bands in the lane labelled C3-2, the top band is the right size, the bottom one is approximately 500bp.  Please discuss what it might be in discussion.

Table 3: please use a period “.” rather than a comma “,” to show decimals.  Please also keep only two decimals in the column labelled “Odds ratio” to be consistent with the mani text.  What do you mean by “Unlettered”, illiterate?

L267-8: “Furthermore, only 14.1% (12/85) of the animals with a positive sublingual cysticercosis inspection had a positive Ag-ELISA test”.  This statement is incorrect.  Eighty-even pigs were visually examined for cysticercus sublingually, 25 were found positive.  The rate should be 48.0% if 12 were found cysticercus Ag positive (12/25) unless I misunderstood what you meant here.

M&M section: use centrifuge force (×g) rather than rpm since rpm is meaningless without the diameter of the rotor used. Also please add information for each supplier such as CellSens, Olympus microscope and DP74 camera

Subtitle for the section 3.4.- considering change it to “Risk factor…”.

Comments on the Quality of English Language

See my comments above. 

Author Response

Dear Reviewer

I greatly appreciate your feedback, which has helped to improve my manuscript significantly. Please find our answer attached and a new manuscript version.

Sincerely

Jenny J Chaparro-Gutierrez
